# The Light and Dark Sides of Virtual Screening: What Is There to Know?

**DOI:** 10.3390/ijms20061375

**Published:** 2019-03-19

**Authors:** Aleix Gimeno, María José Ojeda-Montes, Sarah Tomás-Hernández, Adrià Cereto-Massagué, Raúl Beltrán-Debón, Miquel Mulero, Gerard Pujadas, Santiago Garcia-Vallvé

**Affiliations:** 1Research group in Cheminformatics & Nutrition, Departament de Bioquímica i Biotecnologia, Universitat Rovira i Virgili, Campus de Sescelades, 43007 Tarragona, Catalonia, Spain; aleix.givi2@gmail.com (A.G.); mjoseom88@gmail.com (M.J.O.-M.); sarahtomas89@gmail.com (S.T.-H.); ssorgatem@gmail.com (A.C.-M.); raul.beltran@urv.cat (R.B.-D.); miquel.mulero@urv.cat (M.M.); 2EURECAT, TECNIO, CEICS, Avinguda Universitat, 1, 43204 Reus, Catalonia, Spain

**Keywords:** bioactivity prediction, cheminformatics, drug discovery, medicinal chemistry, virtual screening

## Abstract

Virtual screening consists of using computational tools to predict potentially bioactive compounds from files containing large libraries of small molecules. Virtual screening is becoming increasingly popular in the field of drug discovery as in silico techniques are continuously being developed, improved, and made available. As most of these techniques are easy to use, both private and public organizations apply virtual screening methodologies to save resources in the laboratory. However, it is often the case that the techniques implemented in virtual screening workflows are restricted to those that the research team knows. Moreover, although the software is often easy to use, each methodology has a series of drawbacks that should be avoided so that false results or artifacts are not produced. Here, we review the most common methodologies used in virtual screening workflows in order to both introduce the inexperienced researcher to new methodologies and advise the experienced researcher on how to prevent common mistakes and the improper usage of virtual screening methodologies.

## 1. Virtual Screening

Virtual screening (VS) is a computational technique used to identify from a large library of compounds those that bind to a specific target, usually an enzyme or receptor. Virtual screening is usually approached hierarchically in the form of a workflow, sequentially incorporating different methods, which act as filters that discard undesirable compounds (see Figure 1). This makes it possible to take advantage of strengths and avoid limitations of the individual methods [1,2]. Compounds that survive all the filters of the VS are usually referred to as hit compounds and they need to be tested experimentally in the laboratory to confirm their biological activity. Virtual screening methods can be classified into two major groups: (a) ligand-based methods, which rely on the similarity of the compounds of interest with active compounds, and (b) receptor-based methods, which focus on the complementarity of the compounds of interest with the binding site of the target protein.

Like high-throughput screenings (HTS), VS protocols are normally used as an early step in the drug discovery process in order to enrich the initial library with active compounds [1]. The advantage of VS with respect to HTS is that VS makes it possible to process thousands of compounds in a matter of hours and reduce the number of compounds to be synthesized or purchased and tested, decreasing the costs. In addition, VS can be performed on virtual libraries of compounds, thus expanding the chemical space. A VS procedure does not always allow to obtain compounds with a high activity [1], but its main purpose can be to obtain structurally diverse lead compounds that may be improved in subsequent hit-to-lead and lead optimization stages. In this sense, the results of a VS may, especially if receptor-based methods are used, allow to understand the molecular basis of the activity of active compounds and use this knowledge in the optimization process.

Virtual screening has become an important part of the drug discovery process. Virtual screening methods and VS case studies have been reviewed elsewhere [2,3,4,5,6,7,8,9,10,11]. In this review, we summarize, based on our experience, the most commonly used VS methodologies and discuss their strengths and weaknesses in order to both introduce the inexperienced researcher to new methodologies and advise the experienced researcher on how to prevent common mistakes and the improper usage of VS methodologies.

## 2. First Steps

Before carrying out VS, the available data should be analyzed thoroughly to get to know the target of interest and determine which methodologies can or cannot be incorporated in the VS workflow:
**Bibliographic research.** First, bibliographic research on the receptor is recommended, considering aspects such as its biological function, natural ligands, and catalytic mechanism, as well as its involvement in pathological processes. This information can be found in databases such as UniProt [12] or Brenda [13]. It is also important to review previous attempts to develop compounds that modulate the activity of the receptor of interest and their mechanisms of action, as well as the current challenges that these compounds face and their limitations. In this regard, the analysis of structure–activity relationship (SAR) studies can provide useful insights into how to design inhibitors for a given target. Structure–activity relationship studies are experiments in which a compound is modified by adding a series of different substituents in one or several parts of the molecule and evaluating the experimental activity of the resulting compounds towards the target of interest. This provides information on the compound substituents that are preferred by the target in each part of the molecule, and therefore, reveals which modifications could be applied to a compound to further increase its activity towards the target and which ones should not be applied, as they would result in activity losses. Although in some cases, molecular visualization software could aid us in rationalizing the causes of the activity changes observed in SAR studies by analyzing them within the protein environment with the naked eye, there are methods that help us to better understand the nature of the interactions between the ligand and the protein. For example Flare [14] can be used to represent and compare the electrostatic potentials and hydrophobicity of both the ligand and the protein to determine whether the introduction of a particular functional group is favorable for activity or not. While this type of information may be important for establishing an appropriate VS strategy, it can also be crucial in the final steps of the VS to: (a) select hits for bioactivity testing that show features which have previously been reported to be important for activity; and (b) avoid selecting compounds that would perform unfavorable interactions with the target that may have been overlooked by earlier steps of the VS workflow.**Activity and structural data collection.** On the one hand, activity data of previously reported inhibitors as well as their structure should be retrieved from databases such as ChEMBL [15], Reaxys [16], BindingDB [17] or PubChem [18], as a large amount of structure variability of compounds will improve the performance of the developed ligand-based models and give a more representative computational validation of the VS methods used [1]. On the other hand, it is important to determine whether the 3D structure of the receptor has been elucidated, and if that is the case, the available quantity and quality of crystallographic structures. A careful inspection of these structures will allow us to assess the flexibility of the receptor and evaluate whether receptor-based approaches can be implemented in the VS. Crystallographic models of protein and protein–ligand complexes can be obtained from the Protein Data Bank (PDB) [19,20] database, but it should be kept in mind that the atoms in these models correspond to how the crystallographers have interpreted the data of the electron density maps. Therefore, it is recommended to validate the reliability of such coordinates (especially for those corresponding to the binding site and to the co-crystallized ligand) with specialized visualization software such as VHELIBS [21] before using the PDB files in VS.**Library preparation.** The collection of compounds to which the VS workflow will be applied is referred to as the VS library. Apart from retrieving the structures and activities of compounds with known activity for VS development and validation purposes, the structures of compounds to which the VS should be applied also need to be obtained, generating the so-called virtual screening library. This library of compounds can come from an in-house collection of compounds of interest, it can be obtained from different databases, such as ZINC [22] or Reaxys [16], or directly from a commercial compound supplier. In many cases, the structures of these compounds are collected in 2D format, but many VS methods require the 3D conformation of compounds (i.e., the arrangement of atoms of a molecule in space). Thus, the 3D conformations that molecules adopt usually need to be predicted in a process known as conformational sampling, in which conformers are first generated by determining bond lengths, bond angles, and torsion angles, and then ranked to prioritize the low-energy conformations that are accessible with a reasonable probability at room temperature [23]. The software commonly used to perform these tasks are shown in Table 1. In a recent benchmarking of conformer generator ensembles [24], commercial ensemble generators like OMEGA [25] and ConfGen [26] showed high performance, closely followed by the freely available implementation of the distance geometry [27] algorithm by RDKit [28], which also showed high robustness. In conformer generators, molecules are first fragmented by their rotatable bonds. These fragments are then re-joined while sampling their spatial distributions based on different criteria to obtain conformers. OMEGA [25] and ConfGen [26] are systematic approaches that sample each rotatable bond in the molecule systematically in discrete intervals. The distance geometry [27] algorithm implementation by RDKit [28], on the other hand, is a stochastic method in which the conformational space of a molecule is sampled randomly, but considering a small amount of empirical information [29]. Generating a sufficiently broad set of conformations for each compound is crucial to cover the compound’s conformational space and to achieve optimal results in many VS methodologies that depend on the 3D conformation of compounds (e.g., 3D fingerprints, 3D-shape comparison, electrostatic potential comparison, pharmacophore screening). Otherwise, it is a limitation for subsequent methods if the bioactive conformation of interest is not included among the conformers generated for a given compound [1]. On the other hand, the generation of high-energy conformations that have a low probability of being accessed by the molecule at room temperature should be avoided, as they may be misleading and cause false positive results [1]. In addition to the spatial distribution of atoms, other aspects need to be considered when molecules are prepared for VS purposes. Many VS methodologies are dependent on the charge of molecules (e.g., electrostatic potential comparison, protein–ligand docking, pharmacophore screening), so it is important to ensure that the charges of compounds are properly defined as they may not be present, or they may not be assigned correctly. The different possible protonation states at the pH of interest also need to be generated for each molecule, as well as its tautomeric states [1]. In addition, other aspects such as stereochemistry and the presence of salt and solvent fragments also need to be considered during molecule preparation. Software such as Standardizer [30] or LigPrep [31] and tools such as MolVS [32] can be used for this purpose.

## 3. Ligand-Based Virtual Screening

Ligand-based VS methods measure the similarity of the compounds in the library to reference compounds that are active towards a target of interest or show desired properties. The basis for these methods is the similar property principle introduced by Johnson and Maggiora [48], which states that similar compounds have similar properties. Thus, compounds with high similarity to reference compounds are likely to behave in a similar fashion or act through the same mechanism, and therefore, have similar effects. Similarity is a subjective concept, and different methods use different similarity measures to determine the similarity of two compounds. In this section, the most common ligand-based methods used in virtual screening will be summarized.

Ligand-based VS methods are relatively cheap computationally compared to receptor-based methods, as no macromolecules are involved in the calculations. For this reason they are generally used early in the VS workflow process when the amount of compounds in the starting library is the highest (but could be used also later if no 3D structure is available for the target protein).

### 3.1. Fingerprint-Based Methods

Molecular fingerprints constitute a ligand-based method for similarity searching that uses patterns in the structure of compounds in order to compare them. Fingerprints are sequences of bits, in which each bit includes certain information regarding a molecule (see Figure 2) [49]. As these bits are quantifiable, this makes it possible to draw comparisons between two molecules (A and B) and determine their similarity. According to the nature of the bits, fingerprints can be classified as:**Sub-structure keys-based fingerprints.** In this type of fingerprint, the bit string corresponds to a series of predefined structural keys, and each bit relates to the presence or absence of a given feature in the molecule. Therefore, these fingerprints are effective when the structural keys used by the fingerprint are present in the molecules to be compared, but they are not that meaningful otherwise. Examples of this type of fingerprint include MACCS [50,51], PubChem fingerprints [18], and BCI fingerprints [52].**Topological or path-based fingerprints.** In this type of fingerprint, bits are defined from fragments of the molecules themselves. For every atom in a molecule, fragments are obtained by progressively increasing the length up to a determined number of bonds, usually following a linear path. Then, these fragments are hashed to generate the fingerprint. As fingerprints are generated from the molecules themselves, every molecule produces a meaningful fingerprint, and its length can be adjusted. However, in topological fingerprints with a reduced number of bits, bit collisions may occur as a result of assigning more than one different feature to a given bit. Examples of this type of fingerprint include the Daylight fingerprints [53] and OpenEye’s Tree fingerprints [54].**Circular fingerprints.** In this type of fingerprint, bits are also defined from molecule fragments, but these fragments are obtained from the environment of each atom up to a determined radius instead of a path. Examples of this type of fingerprint include Molprint2D [54,55], extended-connectivity fingerprints (ECFPs), and functional-class fingerprints (FCFPs) [56].**Pharmacophore fingerprints.** A pharmacophore is the spatial arrangement of features that allow the ligands to interact with the binding site of a target protein (see Section 3.4 for more details). Pharmacophore fingerprints incorporate these molecule features and the distances between them into the fingerprint [57].

In addition to the fingerprint types listed above, there are also other fingerprint types, such as fingerprints based on molecule SMILES [58] or protein–ligand interaction fingerprints, which encode information regarding the type of interactions between the protein and the ligand [59,60]. Moreover, fingerprints can also be derived from a combination of the approaches mentioned above, constituting what are known as hybrid fingerprints [61].

Once the fingerprints for each molecule have been calculated, different methods can be used to enrich a library in active compounds through the use of fingerprints:


**(a) Similarity metrics**


Several similarity metrics can be used to compare the fingerprints of two compounds (e.g., Euclidean distance, Manhattan distance or Sørensen–Dice coefficient), but the most popular is the Tanimoto coefficient [62]. The Tanimoto coefficient is a value in a range from 0 to 1 that represents the similarity between two compounds based on the fingerprint bits they match (the higher the value, the more similar the compounds). It is expressed by:
(1)$$\mathrm{Tanimoto}\;\mathrm{coefficient}=\frac{c}{\left(a+b-c\right)}$$
where given the fingerprints of compounds A and B, *a* equals the amount of bits set to 1 in A, *b* equals the amount of bits set to 1 in B, and *c* equals the amount of bits set to 1 in both A and B.

With the help of these similarity metrics, the fingerprints of the compounds in a library can be compared to the fingerprints of active compounds for the target of interest, thus assessing the similarity between them. Next, in order to select the compounds in the library with a higher probability of being active, compounds can be sorted according to the decreasing Tanimoto coefficient or other similarity metrics, and a cutoff can be applied, thus keeping the compounds with a higher similarity to known actives, while discarding compounds with a lower similarity to known actives. While this approach is easy to use and well founded, it has some limitations:A high similarity coefficient value does not always imply that two compounds will have the same activity. Some minor structural changes could greatly modulate the activity of a compound depending on how they affected the interactions with the protein. These great changes in activity due to small changes in compound structure are commonly known as activity cliffs (see Section 4), and they constitute the main reason why activity values cannot always be inferred from similarity measures. Thus, the comparison of similarity coefficients should be seen as an attempt at simplification that bypasses medicinal chemistry in order to establish an automated approximation of the activity of compounds [49,63].There is no universal cutoff value for determining that compounds with a certain range of similarity to reference compounds will have similar activity values. As fingerprints are designed in different ways, if we compare the similarities between two compounds with a given similarity coefficient using two different types of fingerprints, the similarity values obtained will most likely differ. A similar situation will occur when two compounds are compared using the same fingerprint but a different similarity metric [49,63]. Therefore, as the similarity value between two compounds is affected by both the type of fingerprint and the similarity coefficient used, the optimal cutoff will also be dependent on these two factors and will have to be evaluated case by case using the appropriate statistical measures (see Section 5). Moreover, different types of fingerprints perform differently in different situations [1], so the fingerprint results obtained should be validated computationally in order to choose the appropriate fingerprint (see Section 5).Similarity coefficients assign equal importance to all fingerprint bits. This is a limitation in the following two situations: (a) inactive compounds that do not possess the critical features for activity that are present in the active compounds used as a reference but still accomplish good similarity values by matching most of the fingerprint bits will be wrongly predicted to be active; and (b) active compounds that only match the critical features for activity with the active compounds used as a reference and are structurally different from them will be wrongly predicted to be inactive [1,49,63].

Nevertheless, as fingerprint-based similarity approaches display a virtual screening performance similar to other more complex methods while being computationally cheaper, they are still the preferred choice in many VS approaches [49].


**(b) Supervised machine learning**


Another common method for incorporating the information encoded in fingerprints is to use supervised machine learning methods. These consist in different algorithms that, given a sample of the fingerprints of compounds with different activities, can obtain a model that relates the fingerprint to the observed activity value of the compound in order to apply the model to a new set of compounds and predict their activity based on their fingerprints. Some examples of supervised machine-learning methods include random forest [64], support vector machines [65], naive Bayes [66], k-nearest neighbors [67], and artificial neural networks [68]. The application of machine-learning methods to virtual screening for the prediction of activity and pharmacokinetic properties has been extensively reviewed in previous publications [69,70,71,72].

Supervised learning can be divided into classification and regression tasks depending on the desired output. In classification tasks, the objective is to identify which class a particular input belongs to (discrete output). In this case, an arbitrary threshold can be established to divide compounds into active and inactive, and the new compounds will be classified into one of these two categories based on their fingerprints. In regression tasks, the objective is to assign a continuous output value from the input. In this case, the model will be asked to predict the corresponding activity value of a compound with a given fingerprint.

In order to build and validate the model, the input data (in this case, compounds with known activity and fingerprints) need to be divided into training and test sets, both of which should contain active and inactive compounds. While the training set is used to build the model, the test set is used to validate it and evaluate its performance. The different metrics for evaluating model performance are discussed in Section 5. Unlike typical similarity approaches, which are based on the overall similarity of compounds giving equal importance to all parts of the molecule [1], machine-learning methods get around this problem as they are built from multiple active molecules, and fingerprint bits are related independently to the bioactivity of compounds. Thus, they are able to recognize the fingerprint bits that are critical for bioactivity. This constitutes their major strength compared to other ligand-based methods.

### 3.2. 3D-Shape Similarity

In a ligand–receptor interaction, the shape of the ligand is crucial as the ligand needs to fit in the binding pocket of the receptor to establish key interactions for binding to occur. The basis of 3D-shape similarity lies in the fact that two molecules with a similar shape are likely to fit in the same binding pocket and thereby exhibit similar biological activity [73]. In this approach, the 3D shape of the compounds in the VS library is compared to the 3D shape of known active compounds, which are used as a reference. Despite being based on similarity, differently to other ligand-based methods, this method does not take into account the particular structure or properties of the reference ligands and only relies on the shape of the molecules. This is its major advantage and also its major drawback, as it makes it a suitable method for identifying new scaffolds that may overlap well with known ligands which may be active towards the target of interest (see Figure 3), but this does not ensure that the obtained compounds will present the crucial characteristics to exert the desired biological activity. Therefore, 3D-shape similarity is often used in combination with other approaches that account for the chemical properties of compounds. Nevertheless, as new chemotypes are pursued in medicinal chemistry to expand horizons, structural novelty is highly valued in virtual screening and 3D-shape-based similarity analysis is currently gaining more attention in virtual screening campaigns [73]. The software commonly used to perform 3D-shape-based similarity comparisons are shown in Table 1.

Shape comparison methods can be classified into two major categories:**Alignment-free or non-superposition methods.** As these methods are independent of the position and orientations of molecules, they are much faster and could be used to screen large compound databases**Alignment or superposition-based methods.** These methods require a superposition between the reference compounds and the compounds in the database. Although they are highly effective, they are computationally expensive and a sub-optimal alignment may lead to errors in comparing the molecules. These methods make it possible to visualize the alignment together with the similarity values, which can aid in the design of new molecules and be a guide for their further optimization. As an alignment is performed, these methods can also include comparisons of surface properties such as hydrophobicity and polarity.

There are several methods for attaining the common end of evaluating shape similarity. Here is a description of the most commonly used:**Atomic Distance-Based Shape Similarity Methods.** As the shape of a molecule can be described by the relative positions of its atoms, these methods rely on the computation and comparison of inter-atomic distance descriptors to determine shape similarity. These methods do not require the molecules involved to be aligned, and therefore, are faster than alignment-based methods.**Volume-Based Shape Similarity Methods.** Two molecules will have a similar shape if they have a similar volume. Therefore, shape similarity can be described in terms of volume occupancy. The most widely adopted models to describe shape similarity in terms of volume are the hard-sphere model [74,75] and the Gaussian-sphere model [75,76]. The hard-sphere model treats each atom in the molecule as a sphere, and the volume of each molecule is calculated based on the unions and intersections of their volumes. The Gaussian-sphere model represents a molecule as a set of overlapping Gaussian spheres. The inclusion–exclusion principle is applied to obtain the volume of the molecule by calculating the volume of all Gaussians and their intersections.**Surface-Based Shape Similarity Methods.** Shape similarity can also be analyzed by comparing the molecular surfaces of two molecules. Some surface definitions commonly used for this purpose include the solvent-accessible surface [77] and the van der Waals surface [78,79].

Although 3D-shape similarity methods can effectively increase the enrichment in actives of a compound library, compounds with a very similar shape may show different biological behaviors, for instance, due to a low electrostatic complementarity with the binding site or due to the arousal of steric impediments with protein residues. Therefore, it is generally recommended to combine this methodology with other types of methodologies that account for other aspects of the compound or its complementarity with the protein.

### 3.3. Electrostatic Potential Similarity

Another method for measuring the similarity between two compounds is to compare their electrostatic potentials. Electrostatic interactions often play a critical role in the binding of the ligand because the target has a particular electrostatic environment that must be matched by the ligand in order for binding to occur. Thus, using the electrostatic potential of the ligand as a reference, we can obtain compounds that have a similar electrostatic distribution and could potentially match the electrostatic environment of the target, and therefore, they are candidates for having an action on that target. The software commonly used for electrostatic potential comparison is shown in Table 1.

Although structurally similar compounds are likely to have a similar electrostatic potential, some small changes in structure can have a large impact on the electrostatic distribution of the compound (see Figure 4A). Therefore, pairing this methodology with 2D fingerprints or 3D-shape analysis should reduce the number of false positives. More interestingly, compounds with a completely different 2D structure can have similar electrostatic potentials (see Figure 4B). This makes it possible to search for novel inhibitors with the same electrostatic properties as known ligands but with different structures that should be able to bind to the target of interest. When using this methodology to search for IKK-2 and PPARγ inhibitors, Sala et.al. [80] and Guasch et. al. [81] obtained enrichment factors of 4.5 and 11.3, respectively. These results exemplify the performance of this methodology.

### 3.4. Ligand-Based Pharmacophores

According to the IUPAC, a pharmacophore is “the ensemble of steric and electronic features that is necessary to ensure the optimal supramolecular interactions with a specific biological target and to trigger (or block) its biological response” [82]. Pharmacophores which are obtained from one or a set of ligands are called ligand-based pharmacophores. These incorporate the common features throughout a set of ligands that present bioactivity towards a common target. It is then assumed that these features are responsible for the activity of the ligand and the pharmacophore is used to search for other compounds that have the same distribution of features in the VS library (see Figure 5). As the compounds that match the pharmacophore contain the same features as known ligands, they are expected to perform the same interactions with the biological target and their binding is expected to result in the same biological response. The software commonly used in pharmacophore-based virtual screenings is shown in Table 1. The generation of a ligand-based pharmacophore consists of several steps:**Select a set of active ligands to generate the pharmacophore.** First a set of active ligands is obtained, from which the pharmacophore features will be generated, usually based on the common features of the ligands. Ligands that are not active for the target of interest may also be used to discard pharmacophore hypotheses.**Generate 3D conformations of the ligands.** A determined number of low-energy conformations is generated for each bioactive compound so that the conformation in which the ligand binds to the receptor is likely to be included.**Identify ligand features.** The substructures and functional groups of the ligand are transformed to pharmacophoric features. These features often include: (a) hydrogen-bond donor feature, (b) hydrogen-bond acceptor feature, (c) negative-charge feature, (d) positive-charge feature, (e) hydrophobic feature, and (f) aromatic-ring feature. The distances between the features of the ligand are computed and the combination of features and distances is used as an abstract representation of the ligand.**Superimpose ligands.** Ligand representations are superimposed so that a maximum number of features occupy the same regions of space.**Generate pharmacophore.** Features that occupy the same region and are present in the majority of ligands are included in the pharmacophore.**Validation.** The pharmacophore needs to be validated to ensure that it is able to discriminate active compounds from inactive compounds. This is usually done by screening a set of actives and a set of decoys, which are compounds similar to actives that do not present bioactivity for the target of interest (see Section 5).

As the user determines the number of pharmacophore features and their tolerances in the model, it is important to correctly evaluate its strictness. While a very strict model leads to better activity results but poor structural diversity, a very fuzzy model is more likely to retrieve a larger number of false positives but achieve a higher structural diversity. Therefore, an adequate trade-off should be found between strict and loose criteria (see Section 5) [1]. This can be achieved by prioritizing features that show a better performance or that are associated with higher compound activity. For instance, it is possible to develop and evaluate the performance of a particular pharmacophore with and without the presence of certain features or adjust their tolerances in order to determine the importance of each feature for the performance of the model. In order to prioritize pharmacophoric features that are relevant for compound activity, information obtained from SAR studies regarding ligand–receptor interactions that are critical for activity can be included in the pharmacophore model.

Although a pharmacophore model can be developed from a single or a few active molecules, it is recommended to use a rather large set of actives to develop the model based on their common features [1]. This is because using one or a few ligands that do not present certain features that are relevant for activity as the only reference compound/s may lead to certain important features for bioactivity being missed out and a lower enrichment in active compounds after applying the pharmacophore to the VS library.

Pharmacophores are normally used to identify compounds that will present a given biological response. However, compounds with an undesired biological response may also be filtered out by using an anti-pharmacophore, which is a pharmacophore that contains the features that are not desired in a compound. For instance, in a study by Guash et al. [81], the authors identified PPARγ partial agonists of natural origin using an anti-pharmacophore to exclude possible PPARγ full agonists.

## 4. Receptor-Based Virtual Screening

Previously, we have seen how ligand-based VS methods take advantage of the similar property principle in order to identify active compounds towards a particular target. Nevertheless, although similar compounds often have similar activities, this is not always the case, as some compound modifications may be prejudicial for the ligand-target interaction and therefore result in a loss of activity for the target of interest. Thus, this can lead to erroneous predictions if the similar property principle is applied to determine the activity of the new compound [1]. For instance, if a negatively charged carboxylic acid group is introduced in a region of the molecule that is close to an acidic residue of the target protein, such as Glu or Asp, even though the new compound and the original compound will be structurally similar because they have the same substructure, this will most likely result in a loss of activity of the compound for that target due to the electrostatic repulsion of the negative charges in the new compound and the protein. These situations in which a small modification of the compound results in a drastic change of activity are known as activity cliffs. To avoid these types of incompatibilities between a compound and the receptor, it is important to take the receptor into account. Nevertheless, as the receptor is a macromolecule, more information needs to be processed and, thus, receptor-based methods are more computationally expensive than ligand-based methods.

### 4.1. Protein–Ligand Docking

Protein–ligand docking is the most popular structure-based technique in virtual screening [83]. This methodology uses the crystallized structure of a protein to predict how the compounds in the VS library would bind to the binding site (see Figure 6). The different compound orientations in the binding site generated by docking are referred to as docked poses. The software commonly used to perform protein–ligand docking are shown in Table 1. Protein–ligand docking consists of the following steps:
**Protein preparation.** As experimental structures, X-ray crystallographic protein structures present problems such as missing hydrogen atoms, missing residues, incomplete side-chains, undefined protonation states or the presence of crystallization products that are not found *in vivo*. These aspects need to be corrected before the crystallographic structure can be used to perform protein–ligand docking.**Binding site definition.** The limits of the cavity of the protein in which compounds should be docked can be defined to restrict the space occupied by docked poses. In this step it is possible to define constraints to require that the ligands perform certain interactions with the protein or occupy a certain space within the binding site.**Conformational sampling.** A search algorithm is responsible for identifying the possible conformations (docked poses) in which each compound may fit in the binding site. Constraints can be defined during docking to require the resulting docked poses to bind to a certain region of the binding site or to perform a certain interaction with the receptor.**Scoring.** Finally, the affinity of each docked pose for the target is approximated with a scoring function that predicts the strength of the interaction, generating a score for each docked pose. Then, the docked poses are ranked according to the score provided by the docking function to obtain the docked pose that is most likely to represent the real binding mode of the compound.

As docking is one of the most popular and available VS methods, docking results are often misinterpreted by inexperienced researchers for the following reasons:
Although the search algorithm provides potential orientations of the compound in the binding site, this does not imply that the compound can actually bind to the protein. Moreover, even if the compound is an actual ligand of target, this does not imply that the real binding mode of the compound is among the docking poses, as the search algorithm can fail to predict it. Thus, docking should be seen as a means of generating hypotheses on how the compound may bind in the binding site of the target (i.e., docked poses), but not as definite proof that the compound binds in a determined fashion [1,84].Docking software are often wrongly used to predict the activity of compounds based on the score provided by docking functions. Although this may be possible in some cases for structurally similar compounds, scoring functions are not accurate enough to predict the binding affinity of compounds that have different structures and different predicted binding modes. The low success rate of scoring functions at predicting the binding affinity of compounds should be taken into account when a docking simulation is performed [1,84,85]. Instead of aiming at predicting compound activity, protein–ligand docking should be used to enrich the initial library in active compounds by discarding compounds that are not able to fit in the binding site of the protein and by keeping the compounds that are more likely to show a good binding affinity as predicted by the scoring function. The latter can be achieved, for instance, by establishing a docking score threshold.The flexibility of the protein can be accounted for in docking procedures in an approach known as induced-fit docking [86]. This is often not the case in virtual screening, as this methodology implies an added computational cost. Instead, usually only the flexibility of the ligand is considered, and the receptor atoms are not allowed to change their spatial location. However, in proteins with a flexible binding site able to accommodate very diverse ligands, this may not be the right approach and allowing the movement of some protein residues could be considered [1]. A possible workaround to account for the flexibility of the protein without resorting to induced-fit docking may be to use all the available conformations observed for the receptor to perform protein–ligand docking.

Despite these weaknesses, docking screens are often able to achieve hit rates above 10%. Coupled with its low cost, this justifies the usage of this methodology and explains its popularity [85].

### 4.2. Structure-Based Pharmacophores

Pharmacophores can also be obtained taking into account the receptor. Although the process is similar to a ligand-based pharmacophore, the main difference is the way of obtaining the distribution of features. Instead of obtaining the features through the alignment of ligand conformations, the features that will constitute the pharmacophore can be obtained by one of the following processes:
(a)Using conformations of ligands that are co-crystallized with the receptor. In this case, the pharmacophore features are also obtained from active compounds, but as they are co-crystallized with the receptor, their bioactive conformations are already known and there is no need to generate conformations.(b)Using the residues in the binding site to determine pharmacophore features. In this case, the pharmacophore features are not obtained from the ligands, but rather from the receptor. This method makes it possible to compare pharmacophores obtained from different crystallographic structures of the same target.(c)Docking fragments into the binding site of the receptor. In this case, a fragment library is docked, the docked fragments are clustered, and pharmacophore features are generated from these clusters. The nature of the interaction of the cluster of fragments determines the type of feature, and the docking scores of the fragments in the cluster determine the relevance of the feature. This method makes it possible to probe the binding site and identify interactions not considered in the design of previous inhibitors that could potentially be important for activity. An interesting utility for evaluating the potential activity contribution of each pharmacophoric feature in a fragment-based pharmacophore is the E-pharmacophores [87] utility from Schrödinger, which assigns an energetic contribution to each pharmacophore feature based on the docking scores of the fragments.

An added advantage of receptor-based pharmacophores is that the receptor cavity can be considered in order to directly discard compound conformations that would not fit in the binding site. This can be done by introducing excluded volumes in the regions occupied by the protein atoms that cannot be occupied by the compound as they would result in steric impediments with the protein and obstruct the binding.

## 5. Computational Validation

Because virtual screening workflows consist of a series of computational methodologies, ultimately virtual screening hits are predictions that need to be validated both in silico and in vitro or in vivo to prove their correctness. In silico validation is often performed by applying the virtual screening in parallel to a set of active compounds and a set of inactive or decoy compounds:**Actives.** Active compounds (or simply actives) are compounds which have been reported to have a high activity towards the target of interest. Compounds used as a reference to build the different filters that form the virtual screening workflow fall into this category. The exact activity threshold over which a compound is considered to be active is arbitrary, but compounds are often considered as actives if they have IC50, Ki or EC50 activity values around the micromolar and nanomolar range. The higher the threshold selected, the more restrictive the virtual screening. It should be taken into account that even if a compound has been reported to have a certain activity for the target of interest, the action mechanism may differ from the action mechanism that the VS is seeking (e.g., the compound may exert its action by binding to an allosteric site of the protein instead of the catalytic site). The VS should not be able to identify these compounds as actives as their binding mode is different and this would decrease the performance of the VS. Therefore, active compounds with unreported protein-binding modes represent intrinsic limitations of the VS validation [1].**Inactives.** Inactive compounds (or simply inactives) are compounds which have been reported to have a low activity (or no activity) towards the target of interest. Hit compounds that are similar to inactives are considered to have a higher chance of being inactive, and therefore, should be avoided. Analogously to actives, an arbitrary activity threshold under which compounds are believed to be inactive should be predefined. The higher the threshold, the more demanding will the virtual screening be, as it will be necessary to discern between active and inactive compounds with higher accuracy due to the smaller difference in activity between the two groups. PubChem [18] and ChEMBL [15] are databases of chemical compounds that include inactive compounds.**Decoys.** Decoy compounds (or simply decoys) are compounds that resemble active compounds but for which the activity towards the target of interest has not been reported, and since they are likely to be inactive, they are presumed to be so [35]. Decoys are generally obtained by searching for compounds that have similar physical descriptors (e.g., molecular weight, number of rotational bonds, total hydrogen bond donors, total hydrogen bond acceptors, and octanol–water partition coefficient) to active compounds, but that are chemically different from them (which can be determined by fingerprint similarity) [35]. In validation protocols, decoys are usually used instead of inactives due to the low amount of null results reported in the literature and the consequent lack of data on inactive compounds. Similarly to what occurs with active compounds that act through different action modes than the one assessed by the VS, as decoys are putative inactive compounds but their activity for the target of interest has not been determined, a small portion of them may actually be active, and therefore, this also constitutes an intrinsic limitation for assessing VS performance [1]. Decoys can be obtained either directly from databases, such as DUD-E [88], or through the use of tools, such as DecoyFinder [35], which makes it possible to obtain sets of decoys that match the provided sets of active compounds.

Because each VS step essentially behaves as a binary classifier that labels the output compounds as active (positives) or inactive (negatives), once the active and inactive/decoy groups have been defined and the VS has been applied, each compound will fall into one of the following four classes:
True positives. Active compounds that are predicted to be active.True negatives. Inactive compounds that are predicted to be inactive.False positives. Inactive compounds that are predicted to be active.False negatives. Active compounds that are predicted to be inactive.

These classes conform the so-called confusion matrix of the binary classifier (see Table 2) and they can be used to calculate a series of statistical measures which can in turn be used to assess the performance of each VS step:
Sensitivity. Also referred to as recall, hit rate or true positive rate (TPR), this measures the proportion of actual positives that are correctly identified as such:
(2)$$\mathrm{TPR}=\frac{\mathrm{TP}}{P}=\frac{\mathrm{TP}}{\mathrm{TP}+\mathrm{FN}}=1-\mathrm{FNR}$$Specificity. Also referred to as selectivity or true negative rate (TNR), this measures the proportion of actual negatives that are correctly identified as such:(3)$$\mathrm{TNR}=\frac{\mathrm{TN}}{N}=\frac{\mathrm{TN}}{\mathrm{TN}+\mathrm{FP}}=1-\mathrm{FPR}$$Precision. Also referred to as positive predictive value (PPV), this measures the proportion of positive results that correspond to actual positives:(4)$$\mathrm{PPV}=\frac{\mathrm{TP}}{\mathrm{TP}+\mathrm{FP}}$$Negative predictive value (NPV). This measures the proportion of negative results that correspond to actual negatives:(5)$$\mathrm{NPV}=\frac{\mathrm{TN}}{\mathrm{TN}+\mathrm{FN}}$$False negative rate (FNR). Also referred to as miss rate, this measures the proportion of actual positives incorrectly classified as such (it complements sensitivity):(6)$$\mathrm{FNR}=\frac{\mathrm{FN}}{P}=\frac{\mathrm{FN}}{\mathrm{FN}+\mathrm{TP}}=1-\mathrm{TPR}$$Fall-out. Also referred to as false positive rate (FPR), this measures the proportion of actual negatives that are incorrectly classified as such (it complements specificity):(7)$$\mathrm{FPR}=\frac{\mathrm{FP}}{N}=\frac{\mathrm{FP}}{\mathrm{FP}+\mathrm{TN}}=1-\mathrm{TNR}$$Accuracy. This measures the proportion of correctly predicted results among the total number of cases examined:(8)$$\mathrm{ACC}=\frac{\mathrm{TP}+\mathrm{TN}}{P+N}=\frac{\mathrm{TP}+\mathrm{TN}}{\mathrm{TP}+\mathrm{TN}+\mathrm{FP}+\mathrm{FN}}$$F1 score. This is a measure obtained from the harmonic mean of the sensitivity and the precision statistical measures. It considers both measures in order to determine the performance of the classifier:(9)$$F1=2\cdot \frac{\mathrm{PPV}\cdot \mathrm{TPR}}{\mathrm{PPV}+\mathrm{TPR}}=\frac{2\cdot \mathrm{TP}}{2\cdot \mathrm{TP}+\mathrm{FP}+\mathrm{FN}}$$Matthews correlation coefficient (MCC). This is a correlation coefficient between the observed and predicted binary classifications that returns a value between −1 and +1. A coefficient of +1 represents a perfect prediction, a coefficient of 0 indicates that the prediction is no better than a random prediction, and a coefficient of −1 indicates total disagreement between prediction and observation. The MCC is generally regarded as a balanced measure that can be used even if the classes are of very different sizes. It is calculated using the following formula:(10)$$\mathrm{MCC}=\frac{\mathrm{TP}\cdot \mathrm{TN}-\mathrm{FP}\cdot \mathrm{FN}}{\sqrt{\left(\mathrm{TP}+\mathrm{FP}\right)\cdot \left(\mathrm{TP}+\mathrm{FN}\right)\cdot \left(\mathrm{TN}+\mathrm{FP}\right)\cdot \left(\mathrm{TN}+\mathrm{FN}\right)}}$$

In addition to these statistical measures, other methods are also commonly used to assess the performance of binary classifiers:
Enrichment factor (EF). The EF is a measure of how much the sample is enriched with actives after a determined filter or a series of filters is applied. It is calculated as the ratio between the proportion of actives after and before the VS step.
(11)$$\mathrm{EF}=\frac{\frac{a2}{a2+d2}}{\frac{a1}{a1+d1}}$$
in which:a1 = actives before the VS stepa2 = actives after the VS stepd1 = decoys before the VS stepd2 = decoys after the VS step

With the help of these measures, we are able to evaluate not only the overall performance of the model, but also other characteristics, such as its ability to retrieve actives or discard inactives. Depending on the priorities of the particular step of the virtual screening in question, a determined measure can be prioritized over another in order to select the appropriate model in each situation. For instance, at the beginning of the VS, priority may be given to discarding inactive compounds as the number of inactive compounds in the initial library is higher. In this case, specificity would be prioritized over sensitivity. On the other hand, in later stages of the VS workflow, priority may be given to the retrieval of active compounds, as the proportion of active compounds should be higher and compounds that are predicted to be active are also expected to be active based on previous filters. In this case, sensitivity would be prioritized over specificity. 

Based on these statistical measures, model parameters can be tweaked until a more satisfactory result is achieved. For instance, if after applying a determined workflow filter, the resulting number of compounds is considered to be too high, the parameters of that filter can be set to more restrictive parameters in order to decrease the number of inactives that surpass the filter (therefore reducing the number of false positives) at the expense of losing the ability to correctly predict a proportion of actives (therefore increasing the number of false negatives). In this situation, we would be prioritizing the precision of the model over its accuracy. In a different situation, for instance, if the proportion of actives that surpasses the filter is considered to be low (meaning that the model is too restrictive), the parameters of the model can be tweaked to try to increase the proportion of actives that surpass the filter while avoiding the retrieval of decoys in an attempt to find the optimum compromise between sensitivity and specificity.

Overall, these measures make it possible to evaluate different aspects of the model in an objective manner and adjust its parameters according to the user’s preferences in order to obtain the most suitable model for each situation.

## 6. Hit Selection

To demonstrate the usefulness of the VS workflow, the activity of VS hits for the target of interest needs to be determined in vitro (see Figure 1). This is usually done for a sample of the hit compounds. Subjective selection of compounds should be avoided in order to obtain a representative sample that allows an adequate evaluation of VS performance [1,11,83]. Therefore, it is crucial to select compounds that are different from one another, and this can be achieved by grouping the hit compounds according to their structure using clustering. Clustering is an unsupervised learning method in which the algorithm is provided with input data and its goal is to find patterns in the data and divide the input into groups. Fingerprint data can, for instance, be used as input to cluster compounds according to their structures. This makes it possible to: (a) select hit compounds that belong to different clusters to known actives and that are therefore more novel and of greater interest; (b) avoid selecting compounds that cluster together with inactives and therefore have a greater chance of being inactive; and (c) select hit compounds that belong to different clusters to ensure structural diversity. 

There are different clustering methods, such as hierarchical clustering, k-means clustering or HDBSCAN [89], and the choice of the method is generally influenced by the characteristics of the dataset and the limitations of each method. For instance, in k-means clustering the number of clusters is predefined and they are circumferential, whereas in HDBSCAN [89] the minimum cluster size can be modified to alter the number of clusters and it is also indicated for outlier detection. In hierarchical clustering, an arbitrary similarity threshold can be established to define the desired number of clusters.

## 7. Experimental Validation

As previously mentioned, VS hits are compounds which are expected to have an action on the target of interest, but this has to be demonstrated. The activity of the compound for the target of interest can be tested in the laboratory and this is usually achieved by using a commercial enzymatic activity kit or by developing an in-house method for the detection of enzymatic activity and comparing the activities of the target with and without the presence of the compound that was obtained as a hit.

Nevertheless, the result of a single experiment does not always guarantee that the compound has the action of the target itself, as some compounds may give a positive experimental result by interfering with the assay. Unlike a true drug, which inhibits or activates a protein by fitting into its binding site, these compounds give positive experimental results without performing a specific, drug-like interaction with the protein [90]. Compounds that produce such results are referred to as pan-assay interference compounds (or simply PAINS) [90], and they can be classified by their mode of action:Fluorescent or highly colored compounds. These compounds may give false readouts in fluorimetric and colorimetric assays, giving a positive signal even when no protein is present.Compounds that trap the toxic or reactive metals used to synthesize molecules in a screening library or which are used as reagents in assays. These metals give rise to signals that have nothing to do with the compound’s interaction with the protein.Compounds that sequester reactive metal ions necessary for the reaction.Compounds that coat the protein, altering it chemically and affecting its function in an unspecific way without fitting into its binding site.

Based on previous experience, a series of recommendations is given below on how to discern between hits and PAINS:**Identify potential PAINS based on structure.** Most PAINS fall into 16 different categories according to their chemotypes. Therefore, hits that are candidates of being PAINS can be identified by checking whether they have one of these chemotypes [90]. While this is more effectively done by eye, several in silico tools that implement chemical similarity and substructure searches have also been developed for this purpose [38,91,92,93]. Nevertheless, this does not ensure that all PAINS are discarded and experimental testing will ultimately be needed to identify whether hit compounds giving positive experimental results are acting as PAINS or not.**Perform more than one assay.** To have more certainty that a hit compound which gives a positive experimental result is not a PAIN compound, it is advisable to conduct at least one more assay that detects activity with a different readout in order to check whether the compound is interfering with the assay or not. It is also advisable to check the activity of hits against unrelated targets and if the inhibition of the target of interest is competitive to determine whether the binding of the hit compound is specific or not.

Another reason for which a compound may give a false positive experimental result is aggregation. Some compounds form aggregates that adsorb and denature the protein, inhibiting it in an unspecific way [91]. Molecules that act as aggregators can be predicted computationally [94] and detected experimentally using different tests:Aggregates can be observed directly by dynamic light-scattering as they form particles from 50 to 400 nm in diameter [91,95].Inhibition by colloidal aggregates can be significantly attenuated by small amounts of a non-ionic detergent such as Triton-X or Tween-20 [96].Inhibition by colloidal aggregates can also be attenuated by increasing enzyme concentration, whereas this should not affect the inhibition by well-behaved inhibitors if the receptor concentration to Ki ratio is high [96].Inhibition by colloidal aggregates is non-competitive. If the binding of the compound in question is competitive, the compound is unlikely to be an aggregator [91,96].

It is recommended to combine several of these tests to determine with greater confidence whether the compound acts as an aggregator or a true drug-like compound [96].

Overall, although potential PAINS and aggregators can be discarded *in silico*, further experimental tests should be performed upon confirming the activity of a hit compound in order to determine whether the observed activity is a result of the desired interaction of the compound with the protein, or on the contrary, it corresponds to a false positive result as the compound behaves as a PAIN compound or an aggregator. From a VS design and validation perspective, the fact that a substantial number of molecules reported to be active against a target protein are actually PAINS and aggregators that provided false positive experimental results is a limitation [1] that should be taken into account when: (a) compounds are selected as a reference to perform similarity searches; and (b) the false negatives obtained are evaluated in the VS validation, as PAINS and aggregators may fall into this category.

## 8. ADME

Even if a compound is able to specifically bind to the target of interest and its activity is confirmed *in vitro*, this does not imply that it will have the desired effect *in vivo*. First, the compound will need to be properly absorbed by the organism and distributed to the tissue of interest while avoiding being metabolized and excreted. The properties of a compound that have an influence on its in vivo activity through the modulation of one of these stages are commonly referred to as ADME (Absorption, Distribution, Metabolism, and Excretion) properties and they can be used to determine the drug-likeness of the compound, that is, how a compound resembles an actual drug, and therefore, can be processed as such by the organism.

The ADME properties include physical properties of the compound like its solubility and hydrophobicity. Generally, a drug needs to be soluble enough to be carried by the blood stream, but also lipophilic enough to penetrate the lipidic bilayer that composes the cell membrane. As these properties are inherent to the particular structure of the compound, they can be predicted in silico with the help of mathematical algorithms (see Table 1). Other ADME properties such as the skin, gut–blood, and blood–brain barrier permeability of the compound can also be approximated computationally based on the reported data for known drugs.

One of the most critical aspects regarding the effectiveness of an oral drug, apart from its potency, is its absorption and the proportion of the drug that reaches the blood stream, also referred to as bioavailability. If a compound shows high activity towards a target but has low bioavailability, it will not exert the desired effect. The most popular method for predicting the bioavailability of a compound is Lipinski’s rule of 5, which was developed 20 years ago by Lipinski et al. [97] The rule is based on the ADME properties of known drugs, stating that most of the orally active drugs for humans fulfill three of the following four criteria:A maximum of 5 hydrogen bond donors.A maximum of 10 hydrogen bond acceptors.A molecular weight of less than 500 daltons.An octanol–water partition coefficient not greater than 5.

Therefore, compounds in the screening library that fulfill Lipinski’s rule of 5 are more likely to be orally active and can be filtered either at early stages or at the end of the VS. It should be kept in mind that Lipinski’s rule of 5 only applies to oral bioavailability, and that drugs designed for other administration routes usually fall outside the scope of this rule [1].

## 9. Conclusions and Future Perspectives

Virtual screening consists of the sequential application of different methods to reduce the number of chemical compounds from an initial dataset and enriches this dataset with compounds that have some of the characteristics (ideally those responsible for their activity) of known active molecules for a specific target. In this review, we have introduced the most common methodologies used in VS and their pitfalls and advantages. The methods used in a VS depend on the information available regarding the target of interest. If the three-dimensional structure of the target is available, the use of structure-based methods is recommendable. However, ligand-based methods can be also very effective, especially when SAR studies have been reported and activity cliffs have been described. The combination of ligand-based and receptor-based methodologies may allow the identification of compounds that share the critical characteristics for activity present in known active compounds, while taking into account the complementarity of these compounds with the receptor. Moreover, the combination of methods based on different criteria is highly recommendable to discard compounds that may be incorrectly prioritized by a given methodology. Although there are a lot of successful examples of the use of VS procedures for different targets, each target is different and setting up a VS procedure is not straightforward. Initial preparation of the compounds and structures is a crucial step in a VS. For example, structures of protein–ligand complexes must be validated prior to their utilization and the conformer generation and ligand preparation steps can affect the performance of subsequent steps. The computational validation of each VS step is crucial to refine the methods and establish the appropriate thresholds to better differentiate between actives and inactives or decoy molecules. Finally, hit selection must be taken into account prior to the necessary experimental validation of the predicted activity of the final compounds. Selection of novel scaffolds, identification of potential PAINS and ADME or solubility predictions are some of the possible criteria used in hit selection. The experimental validation of the predicted activity of the final hits of a VS may not be the last step, as the active compounds identified may constitute the starting point of hit optimization processes.

With this review we hope to encourage the use of VS, help researchers familiarize themselves with its capabilities, and most importantly raise awareness of common mistakes in order to promote the proper usage of VS techniques. The use of newly developed methods, such as new predictive algorithms based on machine and deep learning, will increase the possible combinations of methods to be used in VS. One of the aspects that has large room for improvement in the future of VS is the prediction of more potent compounds. Thus, hit selection and hit optimization could be part of the same process. Polypharmacology is also a key concept whose importance is increasing. The same methods used in a VS procedure, with some modifications in some cases, can be used to predict all the bioactivities of a compound, in what is known as target fishing. In this case, the objective is to identify the most probable targets of a query molecule. Combining a VS procedure with target fishing methodologies would allow the identification of multitargeted compounds or the prediction of the adverse effects of a given compound and allow a better hit selection that would facilitate the drug discovery process.

## Figures and Tables

**Figure 1 ijms-20-01375-f001:**
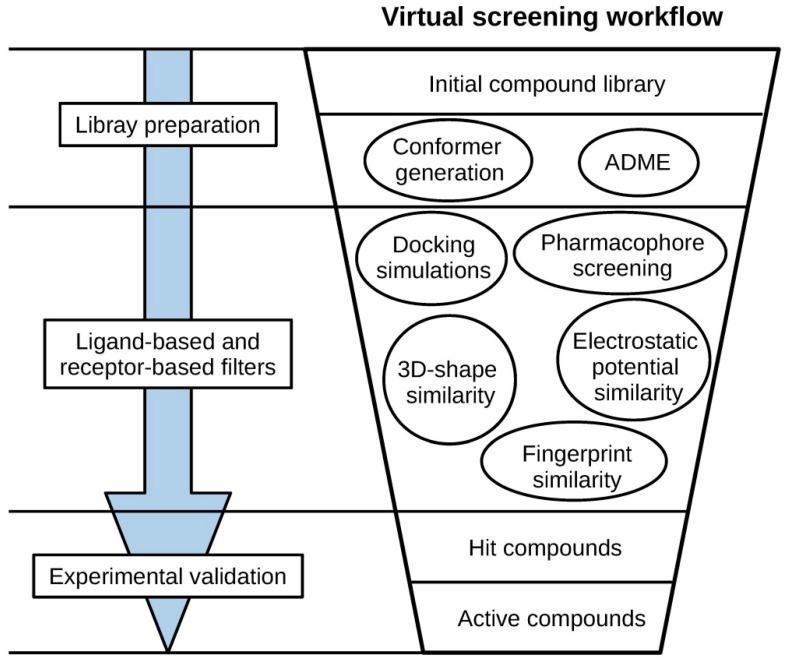
General scheme of a virtual screening workflow.

**Figure 2 ijms-20-01375-f002:**
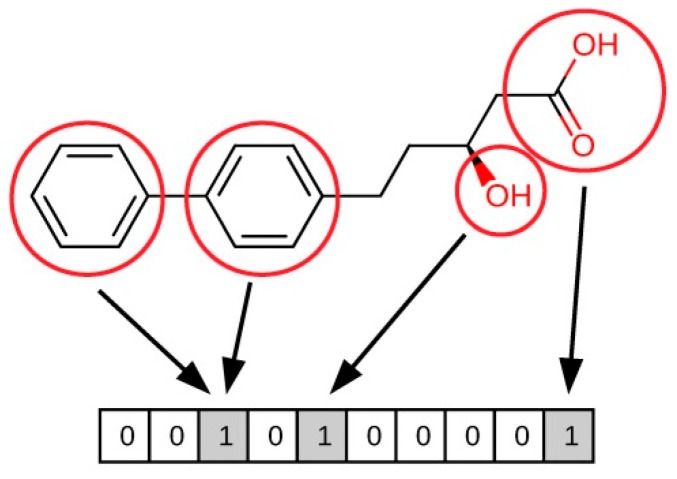
Illustration of how fingerprint bits are derived from the structure of a molecule.

**Figure 3 ijms-20-01375-f003:**
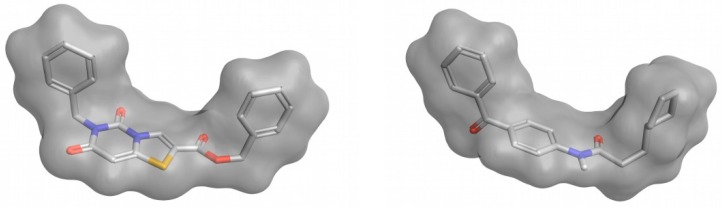
Example of two compounds with a different molecular structure but a high 3D-shape similarity. This figure was obtained with Flare [14].

**Figure 4 ijms-20-01375-f004:**
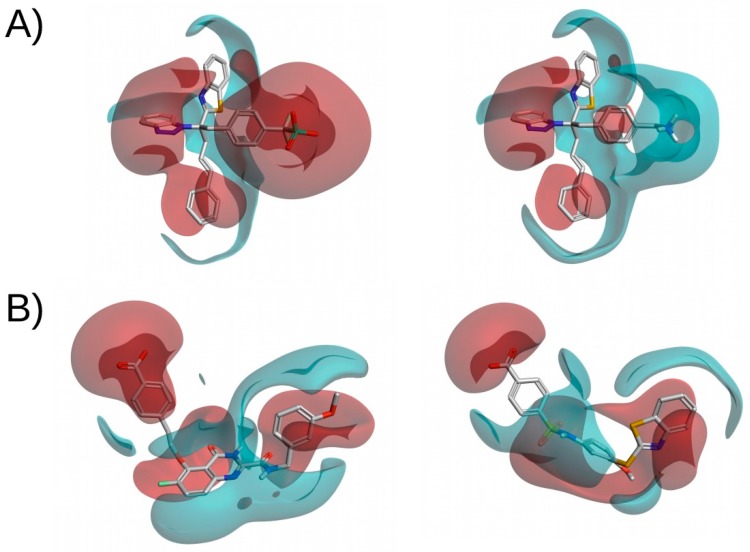
Electrostatic potential comparisons of two pairs of molecules. (**A**) Shows two structurally similar compounds with a different electrostatic potential. (**B**) Shows two compounds with a different molecular structure but a high electrostatic potential similarity. Red and blue surfaces correspond, respectively, to negative and positive electrostatic potentials. This figure was obtained with Flare [14].

**Figure 5 ijms-20-01375-f005:**
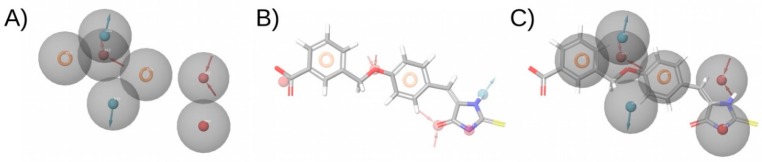
Pharmacophore model and fitting compound. (**A**) shows the pharmacophore model. (**B**) Shows the compound and its pharmacophoric features. (**C**) Shows a superposition of the compound and the pharmacophore, showing that the compound matches four sites in the pharmacophore. The hydrogen bond acceptor, hydrogen bond donor, and aromatic and negative ionizable features are shown in red, blue, and orange and red, respectively. The arrows in the hydrogen bond acceptor and hydrogen bond donor features indicate the direction of the hydrogen bond. This figure was obtained with Maestro [33].

**Figure 6 ijms-20-01375-f006:**
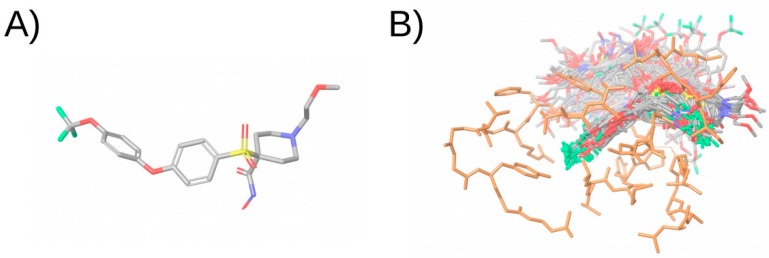
Illustration of the docking simulation of a compound. (**A**) Shows the molecular structure of the compound. (**B**) Shows the docked poses obtained after docking the compound in the binding site of the protein. The compound is colored in the CPK color scheme and the protein is colored in orange. Non-polar hydrogen atoms have been omitted in the representation. This figure was obtained with Maestro [33].

**Table 1 ijms-20-01375-t001:** Popular and useful software used in the different steps involved in a virtual screening workflow. A more complete list can be found at http://www.click2drug.org/.

Method	Software	Developer
Graphical user interface	Flare [14]	Cresset
	Maestro [33]	Schrödinger, LLC
	VIDA [34]	OpenEye Scientific Software Inc.
Decoy set preparation	DecoyFinder [35]	Cheminformatics and Nutrition Research Group (Universitat Rovira I Virgili)
Crystal structure validation	VHELIBS [21]	Cheminformatics and Nutrition Research Group (Universitat Rovira I Virgili)
Molecule standardization	Standardizer [30]	ChemAxon
	LigPrep [31]	Schrödinger, LLC
	MolVS [32]	RDKit
Conformer generation	OMEGA [25]	OpenEye Scientific Software Inc.
	ConfGen [26]	Schrödinger, LLC
	Distance Geometry (DG) [27]	RDKit
	ETKDG [29]	RDKit
ADME property prediction	QikProp [36]	Schrödinger, LLC
	SwissADME [37]	Swiss Institute of Bioinformatics
	FAFDrugs4 [38]	UMRS Paris Diderot-Inserm 973
Shape similarity	ROCS [39]	OpenEye Scientific Software Inc.
	Shape screening [40]	Schrödinger, LLC
Electrostatic potential similarity	EON [41]	OpenEye Scientific Software Inc.
Pharmacophore	Phase [42]	Schrödinger, LLC
	Ligandscout [43]	Inte:Ligand GmbH
Docking	Glide [44]	Schrödinger, LLC
	GOLD [45]	The Cambridge Crystallographic Data Centre
	DOCK [46]	University of California San Francisco
	Autodock [47]	The Scripps Research Institute

**Table 2 ijms-20-01375-t002:** Confusion matrix of a binary classifier.

Predicted Condition	True Condition	
	Positive	Negative
Positive	True positives (TP)	False positives (FP)
Negative	False negatives (FN)	True negatives (TN)

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
