# Peer review of "The Light and Dark Sides of Virtual Screening: What Is There to Know?"

_ijms, 2019, doi:10.3390/ijms20061375_

Round 1
Reviewer 1 Report
The manuscript titles ‘The light and dark sides of virtual screening: what is there to know?’ discusses various strategies to virtual screening. The topics are comprehensively conversed and well-written. The manuscript can still be improved to interest more readership.
1) The figures should be improved.
2) Specific literature examples can be provided to depict the light and dark sides of the different methods.
3) A summary table of the pitfalls, validations, and advantages of each method would benefit to conclude the reviewed literature.
4) There are some similar review articles (Curr Med Chem. 2013;20(23):2839-60). Comment on how this review is different from those already published.
Author Response
Reviewer #1:
"1) The figures should be improved."
We fully agree with the comment of reviewer #1 that figures in the first submission had not enough resolution. We have improved the resolution of all figures and changed a little bit the design of figure 1.
"2) Specific literature examples can be provided to depict the light and dark sides of the different methods.".
We agree with referee's comment that specific literature examples can be used to exemplify some parts of the manuscript. We have introduced therefore the following articles:
· 80. Sala et al. Identification of human IKK-2 inhibitors of natural origin (part I): modeling of the IKK-2 kinase domain, virtual screening and activity assays. PLoS One 2011, 6, e16903.
· 81. Guasch et al. Identification of PPARgamma Partial Agonists of Natural Origin (I): Development of a Virtual Screening Procedure and In Vitro Validation. PLoS One 2012, 7, 1–13.
Thus, at the “3.2. 3D-shape similarity” section we have added the following sentences: “When using this methodology to search for IKK-2 and PPARγ inhibitors, Sala et.al.[80] and Guasch et. al.[81] obtained enrichment factors of 4.5 and 11.3, respectively. These results exemplify the performance of this methodology.”
The following paragraph has also been added at the end of the “3.4. Ligand-based pharmacophores” section: “Pharmacophores are normally used to identify compounds that will present a given biological response. However, compounds with an undesired biological response may also be filtered out by using an antipharmacophore, which is a pharmacophore that contains the features that are not desired in a compound. For instance, in a study by Guash et. al., the authors identified PPARγ partial agonists of natural origin using an antipharmacophore to exclude possible PPARγ full agonists.[81]”
And the following two sentences have been added at the end of the “4.1. Protein-ligand docking” section: “Despite these weaknesses, docking screens are often able to achieve hit rates above 10%. Coupled with its low cost, this justifies the usage of this methodology and explains its popularity.[85]”
In addition, we have added several reviews and articles describing case studies at the end of the Introduction section, and at the “3.1 Fingerprint-Based Methods” section (see below the answers to others reviewer’s comments).
"3) A summary table of the pitfalls, validations, and advantages of each method would benefit to conclude the reviewed literature.".
We agree with the reviewer's opinion that in some cases a table is very useful to summarize some parts of a manuscript. In this case, however, we think that this kind of table would have to introduce too many ideas to fully understand it and for this reason it is better to include the pitfalls and advantages of each method in each section, where each method is introduced and explained.
"4) There are some similar review articles (Curr Med Chem. 2013;20(23):2839-60). Comment on how this review is different from those already published.".
Our reviewer tries to summarize, based on our experience, the VS methodologies. Our target readers are researchers with no experience in this subject, for whom we introduce the most important information for each methodology, and researchers with previous experience in this field, for whom we introduce the pitfalls and advantages of each method. We think that our review is complementary to other reviews (like that the reviewer cites) and for this reason we have added the following paragraph and the end of the Introduction section:
“VS has become an important part of the drug discovery process. VS methods and VS case studies have been reviewed elsewhere. [2-11] In this review we summarize, based on our experience, the most commonly used VS methodologies and discuss their strengths and weaknesses in order to both introduce the inexperienced researcher to new methodologies and advise the experienced researcher on how to prevent common mistakes and the improper usage of VS methodologies.”
Reviewer 2 Report
The manuscript by Gimeno et al. summarizes the main virtual screening techniques commonly applied in the drug discovery field for hit identification. The different computational strategies are described illustrating the concepts they are based on, together with their own strengths and intrinsic limitations. Moreover, the topics of virtual screening validation and final hit selection are taken into account, and useful guidelines generally applicable when employing each of the described techniques are provided.
The manuscript is written and organized in a clear form that makes it easy to read by the non-expert researchers, to which the manuscript is primarily addressed, although some topics could be discussed more thoroughly. I believe that the manuscript can deserve publication after the following minor revisions.
Minor points.
The resolution of all figures appears to be too low and some figures are too small (i.e. figures 1, 4 and 5). I suggest to use figures with higher resolution, in general, and bigger versions of figure 1, 4 and 5).
- The first section of the manuscript (1. Virtual Screening) is rather short and does not properly introduce the topic. I would expect to find here more information about the advantages of using VS protocol for hit identification with respect to HTS or other strategies, possibly citing some reference example in which VS successfully led to important discoveries for the medicinal chemistry field. See as an example the paper by Irwin and Shoichet for successful case studies about docking-based VS campaigns (J Med Chem. 2016, 59, 4103-20)
- Page 3, lines 109-112. After this sentence, few lines about the different algorithms implemented in the software cited should be reported here.
- Page 5, lines 178-180. In this brief description, both the terms "pharmacophore" and "features", are introduced without providing the reader with any type information about what they refer to. I believe that the authors should give here a minimal hint about what is a pharmacophore feature, in order to make the definition of pharmacophore fingerprint comprehensible to the non-experienced reader. Moreover, a link to section 3.4, e.g. "(see section 3.4 for more details)”, should be added in order to indicate where the concept of pharmacophore will be more extensively described.
- Page 6, lines 201-203. Equation (1) is wrong. The Tanimoto coefficient corresponds to c/(a+b-c). Please correct the equation. Moreover, in lines 202-203, the letters “a”, “b” and “c” identifying the variables of equation (1) should be in italics.
- Page 6, lines 235-239. Since it is possible to talk about a “wrong prediction” of activity or inactivity for a certain compound only if we actually know if the compound is endowed with some activity or not (either prior to or after the VS), I would be more specific here. The sentence should be corrected saying that “a) inactive compounds that … will be wrongly predicted to be active; b) active compounds that … will be wrongly predicted to be inactive”. In fact, as an example, there could be compounds presenting the key features necessary for the activity towards a certain target (which are present in the reference actives) that are however inactive because of other structural reasons (e.g. bulky groups that hamper the binding to the target, not present in the active compounds). Such compounds would be probably successfully predicted as inactive (although they present the key binding features) due to the low overall similarity to the reference actives.
- Page 7, lines 251-252. Please provide some references relative to the machine learning methods cited here.
- Page 7, section 3.2 (3D-shape similarity). In this section the general concepts of shape-based similarity methods are illustrated, as well as the different types of shape based alignments. However, although the advantages of these techniques are reported, there is no hint about their possible limitations. For instance, compounds with very similar shape can show very different biological behaviors for many reasons and shape-based (especially pure shape-based) similarity methods can be detrimental for certain types of VS studies (although very good for others). Some considerations about the limitations of shape-based similarity methods should be added to this section.
- Page 11, lines 446-450. Instead of talking about grid definition it is better to talk about binding/docking site definition. Many docking programs do not require the definition of a grid for performing calculations, but only to delimit the receptor binding site.
- Page 11, line 453. Replace “grid” with “binding site”.
- Page 12, lines 472-474. The docked compounds can be completely inactive towards the protein target, but the docking algorithm will always produce a possible binding mode for them. I suggest to rephrase the sentence as follows: "Although the search algorithm provides potential orientations of the compound in the binding site, this does imply that the compound can actually bind to the protein. Moreover, even if the compounds is an actual ligand of target, this does not impliy that the real binding mode of the compound is among the docking poses, as the search algorithm can fail to predict it.”
- Page 12, line 489. The authors misunderstood the definitions of rigid and flexible docking. Rigid docking corresponds to a docking evaluation in which the ligand is maintained rigid: the coordinates of the ligand are rotated and translated during the calculation without altering the molecular conformation. Flexible docking corresponds to the (currently) common approach in which the conformational space of the docked ligand is sampled in the calculation. A docking evaluation in which some selected protein residues included in the docking site are considered as flexible is defined as induced-fit docking, while the use of multiple protein conformation as receptor structures for docking studies is defined as ensemble docking. The paragraph should be revised according to the proper definitions.
- Page 19, Conclusions. The conclusions are way too short. They should summarize the useful guidelines relative the development, application and validation of VS approaches provided by the Authors. Moreover, the conclusions should report some final remark about the use of VS in hit finding and possible future perspectives related to these methodologies.
There are also a series of grammar/typing mistakes throughout the manuscript that should be corrected. I report here some examples.
- Page 5, line 149. Replace “highest” with “the highest”.
- Page 6, line 277. Replace “different” with “differently”.
- The terms “in silico”, “in vitro”, “in vivo” should be in italics.
Author Response
Reviewer #2:
"The resolution of all figures appears to be too low and some figures are too small (i.e. figures 1, 4 and 5). I suggest to use figures with higher resolution, in general, and bigger versions of figure 1, 4 and 5).".
We fully agree with the comment of reviewer #2 that figures in the first submission had not enough resolution. We have therefore improved the resolution of all figures. In addition, following the reviewer’s comment we have enlarged figures 1, 4 and 5. See also response 1 to reviewer #1.
"The first section of the manuscript (1. Virtual Screening) is rather short and does not properly introduce the topic. I would expect to find here more information about the advantages of using VS protocol for hit identification with respect to HTS or other strategies, possibly citing some reference example in which VS successfully led to important discoveries for the medicinal chemistry field. See as an example the paper by Irwin and Shoichet for successful case studies about docking-based VS campaigns (J Med Chem. 2016, 59, 4103-20)".
We are grateful to the reviewer for the suggestion. We have changed some parts of the introduction and we have added the following text at the end of the introduction, where we comment now the advantages of using VS against HTS and citing other reviews and case studies:
“The advantage of VS with respect to HTS is that VS makes it possible to process thousands of compounds in a matter of hours and reduce the number of compounds to be synthesized or purchased and tested, decreasing the costs. In addition, VS can be performed on virtual libraries of compounds, thus expanding the chemical space. A VS procedure does not always allow to obtain compounds with a high activity,[1] but its main purpose can be to obtain structurally diverse lead compounds that may be improved in subsequent hit-to-lead and lead optimization stages. In this sense, the results of a VS may, especially if receptor-based methods are used, allow to understand the molecular basis of the activity of active compounds and use this knowledge in the optimization process.
VS has become an important part of the drug discovery process. VS methods and VS case studies have been reviewed elsewhere. [2–11] In this review we summarize, based on our experience, the most commonly used VS methodologies and discuss their strengths and weaknesses in order to both introduce the inexperienced researcher to new methodologies and advise the experienced researcher on how to prevent common mistakes and the improper usage of VS methodologies.”
"Page 3, lines 109-112. After this sentence, few lines about the different algorithms implemented in the software cited should be reported here.".
We fully agree with the comment of reviewer #1 that some information about the different algorithms should be reported. For this reason we have added the following paragraph at the “Ligand Preparation” section:
“In conformer generators, molecules are first fragmented by their rotatable bonds. These fragments are then re-joined while sampling their spatial distributions based on different criteria to obtain conformers. OMEGA[25] and ConfGen[26] are systematic approaches that sample each rotatable bond in the molecule systematically in discrete intervals. The Distance Geometry[27] algorithm implementation by RDKit,[28] on the other hand, is a stochastic method in which the conformational space of a molecule is sampled randomly, but considering a small amount of empirical information.[29]”
"Page 5, lines 178-180. In this brief description, both the terms "pharmacophore" and "features", are introduced without providing the reader with any type information about what they refer to. I believe that the authors should give here a minimal hint about what is a pharmacophore feature, in order to make the definition of pharmacophore fingerprint comprehensible to the non-experienced reader. Moreover, a link to section 3.4, e.g. "(see section 3.4 for more details)”, should be added in order to indicate where the concept of pharmacophore will be more extensively described.".
Following the reviewer’s suggestion, we have added the following definition of the term pharmacophore and a link to section 3.4, where the concept of pharmacophore is more extensively described:
“A pharmacophore is the spatial arrangement of features that allow the ligands to interact with the binding site of a target protein (see section 3.4 for more details). Pharmacophore fingerprints incorporate these molecule features and the distances between them into the fingerprint.[57]”
"Page 6, lines 201-203. Equation (1) is wrong. The Tanimoto coefficient corresponds to c/(a+b-c). Please correct the equation. Moreover, in lines 202-203, the letters “a”, “b” and “c” identifying the variables of equation (1) should be in italics.".
We are grateful to the reviewer for pointing out this error. We have corrected equation (1) and put the letters “a”, “b” and “c” in italics.
"Page 6, lines 235-239. Since it is possible to talk about a “wrong prediction” of activity or inactivity for a certain compound only if we actually know if the compound is endowed with some activity or not (either prior to or after the VS), I would be more specific here. The sentence should be corrected saying that “a) inactive compounds that … will be wrongly predicted to be active; b) active compounds that … will be wrongly predicted to be inactive”. In fact, as an example, there could be compounds presenting the key features necessary for the activity towards a certain target (which are present in the reference actives) that are however inactive because of other structural reasons (e.g. bulky groups that hamper the binding to the target, not present in the active compounds). Such compounds would be probably successfully predicted as inactive (although they present the key binding features) due to the low overall similarity to the reference actives.".
Following the reviewer’s suggestion, we have changed both sentences.
"Page 7, lines 251-252. Please provide some references relative to the machine learning methods cited here.".
Following the reviewer’s suggestion, we have added some bibliographic references relative to the machine learning methods cited in the text (see below) and some references about the application of machine learning methods to VS.
“Some examples of supervised machine learning methods include random forest,[64] support vector machines,[65] naive Bayes,[66] k-nearest neighbors[67] and artificial neural networks.[68] The application of machine learning methods to virtual screening for the prediction of activity and pharmacokinetic properties has been extensively reviewed in previous publications.[69–72]”
"Page 7, section 3.2 (3D-shape similarity). In this section the general concepts of shape-based similarity methods are illustrated, as well as the different types of shape based alignments. However, although the advantages of these techniques are reported, there is no hint about their possible limitations. For instance, compounds with very similar shape can show very different biological behaviors for many reasons and shape-based (especially pure shape-based) similarity methods can be detrimental for certain types of VS studies (although very good for others). Some considerations about the limitations of shape-based similarity methods should be added to this section.".
Following the reviewer’s suggestion, we have changed the “3.2. 3D-shape similarity” section. We have added the following paragraphs about the limitations of shape-based similarity methods.
“…. but this does not ensure that the obtained compounds will present the crucial characteristics to exert the desired biological activity. Therefore, 3D-shape similarity is often used in combination with other approaches that account for the chemical properties of compounds.”
“Although 3D-shape similarity methods can effectively increase the enrichment in actives of a compound library, compounds with a very similar shape may show different biological behaviors, for instance due to a low electrostatic complementarity with the binding site or due to the arousal of steric impediments with protein residues. Therefore, it is generally recommended to combine this methodology with other types of methodologies that account for other aspects of the compound or its complementarity with the protein.”
"Page 11, lines 446-450. Instead of talking about grid definition it is better to talk about binding/docking site definition. Many docking programs do not require the definition of a grid for performing calculations, but only to delimit the receptor binding site.".
Following the reviewer’s suggestion, instead of talking about grid definition, we talk about binding-site definition:
“Binding site definition. The limits of the cavity of the protein in which compounds should be docked can be defined to restrict the space occupied by docked poses. In this step it is possible to define constraints to require that the ligands perform certain interactions with the protein or occupy a certain space within the binding site.”
"Page 11, line 453. Replace “grid” with “binding site”.".
Following the reviewer’s suggestion, we have replaced “grid” with “binding site”.
"Page 12, lines 472-474. The docked compounds can be completely inactive towards the protein target, but the docking algorithm will always produce a possible binding mode for them. I suggest to rephrase the sentence as follows: "Although the search algorithm provides potential orientations of the compound in the binding site, this does imply that the compound can actually bind to the protein. Moreover, even if the compounds is an actual ligand of target, this does not impliy that the real binding mode of the compound is among the docking poses, as the search algorithm can fail to predict it.”.".
We agree with the reviewer’s comment and we have rephrased the sentence as suggested:
“Although the search algorithm provides potential orientations of the compound in the binding site, this does not imply that the compound can actually bind to the protein. Moreover, even if the compound is an actual ligand of target, this does not imply that the real binding mode of the compound is among the docking poses, as the search algorithm can fail to predict it.”
"Page 12, line 489. The authors misunderstood the definitions of rigid and flexible docking. Rigid docking corresponds to a docking evaluation in which the ligand is maintained rigid: the coordinates of the ligand are rotated and translated during the calculation without altering the molecular conformation. Flexible docking corresponds to the (currently) common approach in which the conformational space of the docked ligand is sampled in the calculation. A docking evaluation in which some selected protein residues included in the docking site are considered as flexible is defined as induced-fit docking, while the use of multiple protein conformation as receptor structures for docking studies is defined as ensemble docking. The paragraph should be revised according to the proper definitions.".
We are grateful to the reviewer for noting us this misunderstanding. We have revised the whole paragraph and now we talk about induce fit docking as follow:
“The flexibility of the protein can be accounted for in docking procedures in an approach known as induced fit docking.[86] This is often not the case in virtual screening, as this methodology implies an added computational cost. Instead, usually only the flexibility of the ligand is considered, and the receptor atoms are not allowed to change their spatial location. However, in proteins with a flexible binding site able to accommodate very diverse ligands, this may not be the right approach and allowing the movement of some protein residues could be considered.[1] A possible workaround to account for the flexibility of the protein without resorting to induced fit docking may be to use all the available conformations observed for the receptor to perform protein-ligand docking.”
"Page 19, Conclusions. The conclusions are way too short. They should summarize the useful guidelines relative the development, application and validation of VS approaches provided by the Authors. Moreover, the conclusions should report some final remark about the use of VS in hit finding and possible future perspectives related to these methodologies.".
Following the reviewer’s suggestion, we have changed completely the conclusion section, now entitled “Conclusions and future perspectives”. The new section is as follow:
“VS consists on the sequential application of different methods to reduce the number of chemical compounds from an initial dataset and enrich this dataset with compounds that have some of the characteristics (ideally those responsible for their activity) of known active molecules for a specific target. In this review, we have introduced the most common methodologies used in VS and their pitfalls and advantages. The methods used in a VS depend on the information available regarding the target of interest. If the three dimensional structure of the target is available, the use of structure-based methods is recommendable. However, ligand-based methods can be also very effective, especially when SAR studies have been reported and activity cliffs have been described. The combination of ligand-based and receptor-based methodologies may allow the identification of compounds that share the critical characteristics for activity present in known active compounds while taking into account the complementarity of these compounds with the receptor. Moreover, the combination of methods based on different criteria is highly recommendable to discard compounds that may be incorrectly prioritized by a given methodology. Although there are a lot of successful examples of the use of VS procedures for different targets, each target is different and setting up a VS procedure is not straightforward. Initial preparation of the compounds and structures is a crucial step in a VS. For example, structures of protein-ligand complexes must be validated prior their utilization and the conformer generation and ligand preparation steps can affect the performance of subsequent steps. The computational validation of each VS step is crucial to refine the methods and establish the appropriate thresholds to better differentiate between actives and inactives or decoy molecules. Finally, hit selection must be taken into account prior the necessary experimental validation of the predicted activity of the final compounds. Selection of novel scaffolds, identification of potential PAINS and ADME or solubility predictions are some of the possible criteria used in hit selection. The experimental validation of the predicted activity of the final hits of a VS may not be the last step, as the active compounds identified may constitute the starting point of hit optimization processes.
With this review we hope to encourage the use of VS, help researchers familiarize themselves with its capabilities and, most importantly, raise awareness of common mistakes in order to promote the proper usage of VS techniques. The use of new developed methods, such as new predictive algorithms based on machine and deep learning, will increase the possible combinations of methods to be used in VS. One of the aspects that have a large room for improvement for the future of VS is the prediction of more potent compounds. Thus, hit selection and hit optimization could be part of the same process. Polypharmacology is also a key concept whose importance is increasing. The same methods used in a VS procedure, with some modifications in some cases, can be used to predict all the bioactivities of a compound, in what is known as target fishing. In this case, the objective is to identify the most probable targets of a query molecule. Combining a VS procedure with target fishing methodologies would allow the identification of multitargeted compounds or the prediction of the adverse effects of a given compound and allow a better hit selection that would facilitate the drug discovery process. “
"Page 5, line 149. Replace “highest” with “the highest”.".
Following the reviewer’s suggestion, we have replaced “highest” with “the highest”.
"Page 6, line 277. Replace “different” with “differently”.".
Following the reviewer’s suggestion, we have replaced “different” with “differently”.
"The terms “in silico”, “in vitro”, “in vivo” should be in italics.".
Following the reviewer’s suggestion, we have changed the above terms from all the manuscript and now they are in italics.